# Boron and Phosphorus Co-Doped Graphitic Carbon Nitride Cooperate with Bu$_4$NBr as Binary Heterogeneous Catalysts for the Cycloaddition of CO$_2$ to Epoxides

**Chaokun Yang \***[ID]**, Xin Zhao and Tuantuan Yang**

Discipline Construction Office, Civil Aviation Flight University of China, Guanghan 618307, China
* Correspondence: yangchaokun_hit@163.com

**Abstract:** The development of a cost-effective heterogeneous catalytic system for the cycloaddition reaction of CO$_2$ and epoxides is of great importance. In this manuscript, three kinds of boron and phosphorus co-doping graphitic carbon nitride (BP-CN) were prepared and characterized. Among them, BP-CN-1 displayed the optimal catalytic performance in the presence of Bu$_4$NBr (tetrabutylammonium bromide) for the CO$_2$ cycloaddition with propylene oxide, and 95% propylene carbonate yield was obtained under a 120 °C, 2 MPa, 6 h condition. Moreover, the BP-CN-1/Bu$_4$NBr catalytic system is compatible with various epoxides and also exhibits excellent recycling performance under metal- and solvent-free conditions. Hence, BP-CN-1 exhibited an attractive application for the efficient fixation of CO$_2$ due to the simple, eco-friendly synthesis route and effective catalytic activity.

**Keywords:** graphitic carbon nitride; boron and phosphorus co-doped; CO$_2$ cycloaddition; epoxides; Bu$_4$NBr





## 1. Introduction

In recent decades, CO$_2$ has been considered the primary culprit of the greenhouse effect, which causes global warming, rising sea levels and a series of ecological and environmental problems. CO$_2$ is also deemed an abundant, safe, and renewable C1 building block. Hence, the increasing focus of CO$_2$ research is its environmental friendliness and overall sustainability. In terms of current development, CO$_2$ is a candidate for transformation into high-value-added chemicals by CO$_2$ capture and utilization strategies [1]. Commercial chemicals mainly include urea, methanol, formic acid and organic carbonate, with CO$_2$ as a raw material [2,3]. One approach to CO$_2$ fixation is to synthesize cycle carbonate, which has received considerable attention due to the perspectives of "green chemistry" and "atom economy" [1]. This technology not only eliminates atmospheric CO$_2$ concentrations to a certain extent but also provides a green route to producing cyclic carbonates, which are applied as electrolytes in lithium-ion batteries, pharmaceutical intermediates, and aprotic solvents [4,5]. However, industrialization remains a challenge due to the kinetic inertness and thermodynamic stability of CO$_2$ [6]. To date, a plethora of catalysts that facilitate the process of CO$_2$ cycloaddition have been developed and researched. Homogeneous catalysts and heterogeneous catalysts such as metal complex [7–9], porphyrin [10], metal-organic frameworks (MOFs) [11,12], covalent organic frameworks (COFs) [1], silicon-based material catalysts [13] and carbon nitride (CN) [14] were exploited for the CO$_2$ coupling reaction.

CN, a two-dimensional (2D) layered material, is widely studied due to the metal-free, nontoxic, high chemical, thermal stability and adjustable edge functional groups in the structure, which can be synthesized by the calcination of inexpensive precursors, for instance, urea, melamine, and dicyandiamide [15]. CN possesses excellent structural stability, but CN is mostly applied in the photocatalysis field. Moreover, CN provides rich basic sites, which can be used in specific functional groups to catalyze the reaction to broaden its application field [16,17]. Nitrogen-rich structures in CN are CO$_2$-phillic structures,

guaranteeing high catalytic activity in the cycloaddition of $CO_2$ to epoxides [18]. Many CN catalysts were reported, such as $Zn^{2+}$-doped g-CN/SBA-15 [19], $\gamma$-$Al_2O_3$-supported $ZnBr_2$ and carbon nitride heterogeneous catalysts (Zn–CN/$\gamma$-$Al_2O_3$) [18], $ZnX_2$/CN [20]. Nevertheless, the above catalysts cannot avoid the leaching of metal elements, which is detrimental to the environment. To overcome this hitch, metal-free CN catalysts were explored for $CO_2$ cycloaddition reactions.

Boron-doped carbon nitrides with abundantly exposed edge defects were developed using 1-butyl-3-methylimidazolium tetrafluoroborate ($BmimBF_4$) to enhance the catalytic performance at 130 °C within 6 h [21]. In addition, boron-doped carbon nitride (B-CN), supported on mesoporous silica SBA-15, was used for $CO_2$ cycloadditions under 130 °C, 3 MPa, 24 h conditions. The catalytic activity improvements benefited from the enhancement of surface acid and basic sites in CN [22]. Moreover, a series of phosphorus (P)-doped graphitic carbon nitrides (P-CN) showed excellent performance in $CO_2$ cycloaddition with the presence of $Bu_4NBr$ under mild conditions (100 °C, 2 MPa, 4 h), which proved that the introduction of P could change the electronic structure of CN [23].

Hence, three kinds of boron and phosphorus co-doping graphitic carbon nitride (BP-CN-n, n = 1, 2, 3) materials were prepared and thoroughly characterized for the cycloaddition of $CO_2$ under metal- and solvent-free conditions. Furthermore, the influence of B doping on the cycloaddition reaction was investigated. The effects of reaction time, reaction temperature and $CO_2$ pressure on the yield of cyclic carbonates were discussed. Additionally, the reusability and catalytic reactivity to other substituted epoxides over BP-CN-1 were also investigated.

## 2. Experimental Section

### 2.1. General Information

Boric acid (AR) was provided by Sinopharm Chemical Reagent Co., Ltd. (Shanghai, China). Phosphoric acid (85.0%) was purchased from Tianjin Guangfu Technology Development Co., Ltd. (Tianjin, China). $CO_2$ (99.99%) was obtained from Deyang Shengyuan Gas Co., Ltd. (Deyang, China). Ethyl acetate was supplied from Shanghai Titan Scientific Co., Ltd. (Shanghai, China). Tetrabutylammonium bromide ($Bu_4NBr$), melamine (MA) and all epoxides (99%) were purchased from Aldrich Chemical Co., Ltd. (Shanghai, China). and directly used without further purification.

### 2.2. Synthesis of BP-CN Catalyst

Three kinds of B and P co-doping graphitic carbon nitride (BP-CN) catalysts were synthesized with some improvements based on previous reports [24]. First, MA (5 g, 40 mmol) and phosphorous acid (10 g, 102 mmol) with different quantities of boric acid were dissolved in 100 mL deionized water with vigorous stirring for 30 min. Subsequently, the solution was transferred for hydrothermal processes into a Teflon-lined autoclave and heated to 180 °C for 10 h. Next, the mixture was washed several times with distilled water to remove as much of the remaining $H^+$ as possible meanwhile, the hydrogen phosphite, MA salt, and unreacted boric acid were washed and then filtered and dried at 60 °C overnight. Finally, the white solid was obtained, transferred to a nitrogen oven, and heated at 500 °C for 2 h with a heating rate of 5 °C $min^{-1}$. The resulting samples were designated as BP-CN-1, BP-CN-2 and BP-CN-3 (1.0 g, 16 mmol; 2.0 g, 32 mmol and 3.0 g, 48 mmol boric acid added, respectively). Additionally, pure CN was prepared with MA in a nitrogen oven and labeled as CN. For comparison with BP-CN, P-CN was prepared without boric acid and B-CN was synthesized by replacing phosphorous acid with HCl (4 g, 36.5%, 40 mmol).

### 2.3. Characterizations

Fourier-transformed infrared (FT-IR) spectra were recorded on PerkinElmer Spectrum 100 FT-IR spectrometer with KBr pellet, using the range of 400–4000 $cm^{-1}$. X-ray diffraction (XRD) patterns of the samples were obtained on a Bruker D8 Advance X-ray diffraction with Cu K$\alpha$ radiation ($\lambda$ = 1.5406 Å, 40 kV, 30 mA) irradiation. X-ray photoelectron

spectroscopy (XPS) measurements were performed using a Thermo Fisher Scientic Escalab 250Xi. $N_2$ adsorption–desorption isotherms were measured at liquid nitrogen temperature on Micromeritics ASAP2020 after the catalyst was degassed under vacuum at 180 °C for 10 h. Inductively coupled plasma–atomic emission spectroscopy (ICP-AES) analysis was carried out on ICPOES Optima 8300 from PerkinElmer. High-resolution transmission electron microscopy (HRTEM) images were obtained using a Jem F30 (FETEM, Tecnai G2 F30) with an operating voltage of 200 kV. Scanning electron microscope (SEM) images were recorded on a Hitachi S-8010 scanning electron microscope. Gas chromatography (GC) was analyzed on Agilent GC-7890A instrument with a capillary column (Agilent 19091J-413).

### *2.4. Catalytic Activity Test*

The cycloaddition reaction was conducted in a 25 mL stainless-steel autoclave with an inner Teflon lining. Typically, 0.60 mmol $Bu_4NBr$, 0.100 g BP-CN catalyst and 34.5 mmol propylene oxide were successively added to the autoclave with strong stirring. Then, the reactor was heated to the desired temperature and 2 MPa $CO_2$ was injected into an oil bath for a designated period of time. After the reaction, the autoclave was allowed to cool in an ice-water bath and excessive $CO_2$ was slowly vented. Ultimately, the products were collected by dilution into 20 mL ethyl acetate and analyzed by GC. Meanwhile, the catalyst was recovered by centrifugation and washed with ethyl acetate (5 × 4 mL) to remove the residues. The solid catalyst was collected and dried in the 60 °C oven for the next run. Moreover, $Bu_4NBr$ was partially dissolved in ethyl acetate and the influence of catalyst was mainly explored, so $Bu_4NBr$ was washed in each cycle and new $Bu_4NBr$ was added in the next cycle test.

GC test: In this experiment, the external standard method is used to determine the product yield and selectivity of the reaction. PC is considered the standard sample to test the relationship between its concentration and peak area because of the low boiling point of PO (Figure 1). For other epoxides, epoxides are selected as test samples due to their high boiling points to obtain the relationship between their concentration and peak area. Therefore, the gas phase data provided the PC yield and epoxides conversion. In addition, the detail of catalyst recycling was provided in the "Catalytic activity test". Moreover, the selectivity was calculated on the ratio of the peak area of PC and the total peak areas of all products. GC–MS was adopted to confirm the structure of products of $CO_2$ cycloaddition reaction (Figure S1).

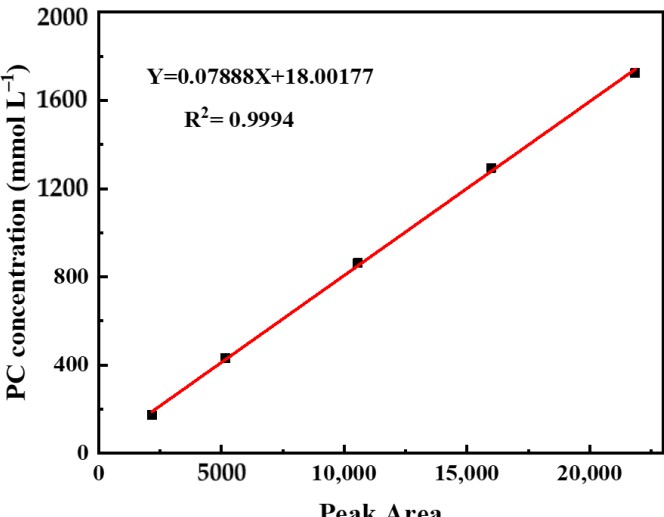

**Figure 1.** Standard curve of peak area versus PC concentration.

## 3. Results and Discussion

### 3.1. Catalyst Characterization

Fourier transform infrared spectrometer (FT-IR) spectra and X-ray diffraction (XRD) patterns were employed to analyze the chemical structures of the synthesized catalysts and the results were presented in Figure 2. As shown in Figure 2A, the broad bands changed from 3000 to 3300 cm$^{-1}$, indicative of primary and secondary amines ascribed to the edges of graphitic CN sheets. The band peaks at 1200–1600 cm$^{-1}$ were attributed to C=N and C–N stretching vibrations of the tri-s-triazine ring. Moreover, the sharp peaks with strong intensity at 810 cm$^{-1}$ were assigned to the presence of triazine units [25,26]. In Figure 2B, the two distinct diffraction peaks at 13.1° and 27.5° corresponded to the (100) plane the in-plane structural packing motif and the (002) plane characteristic of interplanar stacking structures, respectively [27–29]. With the B content increasing, there was no obvious change that suggested that the original structure of CN was well-preserved compared with pure CN.

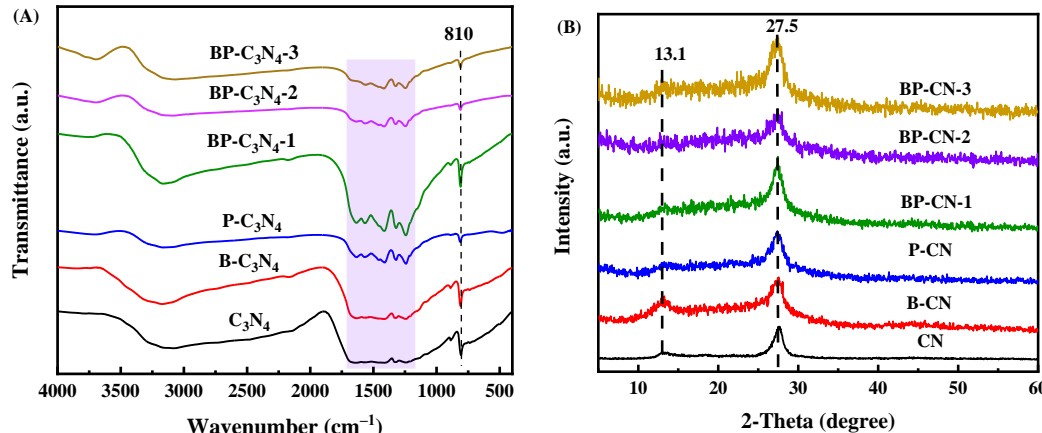

**Figure 2.** (**A**) FT-IR spectra, (**B**) XRD patterns of synthesized catalysts.

X-ray photoelectron spectroscopy (XPS) was adopted to investigate the surface photo-electron state of the elements in BP-CN-1 and the results were exhibited in Figure 3. In Figure 3A, the C 1s spectra were deconvoluted into two peaks at 284.6 eV and 288.0 eV, which were ascribed to C–C and sp$^2$ N–C=N bonds in tri-s-triazine structural units [25,30]. In Figure 3B, N 1s spectra could fit into three peaks: the peak at 398.5 eV was assigned to C–N=C in triazine or heptazine rings, the peak at 399.6 eV resulted from sp$^2$ hybridized N, and 401.1 eV was the binding energy of sp$^3$ terminal N [31,32]. In Figure 3C, the B 1s spectra could fit into two peaks: the 191.6 eV binding energy could be ascribed to the B atoms in the BCN network and the 192.6 eV binding energy could be ascribed to the B atoms surrounded by one N atom and two O atoms [22,33]. Figure 3D illustrated the P 2p spectra of BP-CN-1. The 133.6 eV binding energy could be ascribed to the P–N bond, indicating that P likely replaces C in triazine rings to form P–N bonds [24]. Additionally, the 134.4 eV binding energy belonged to the P–OH of phosphoric acid [23].

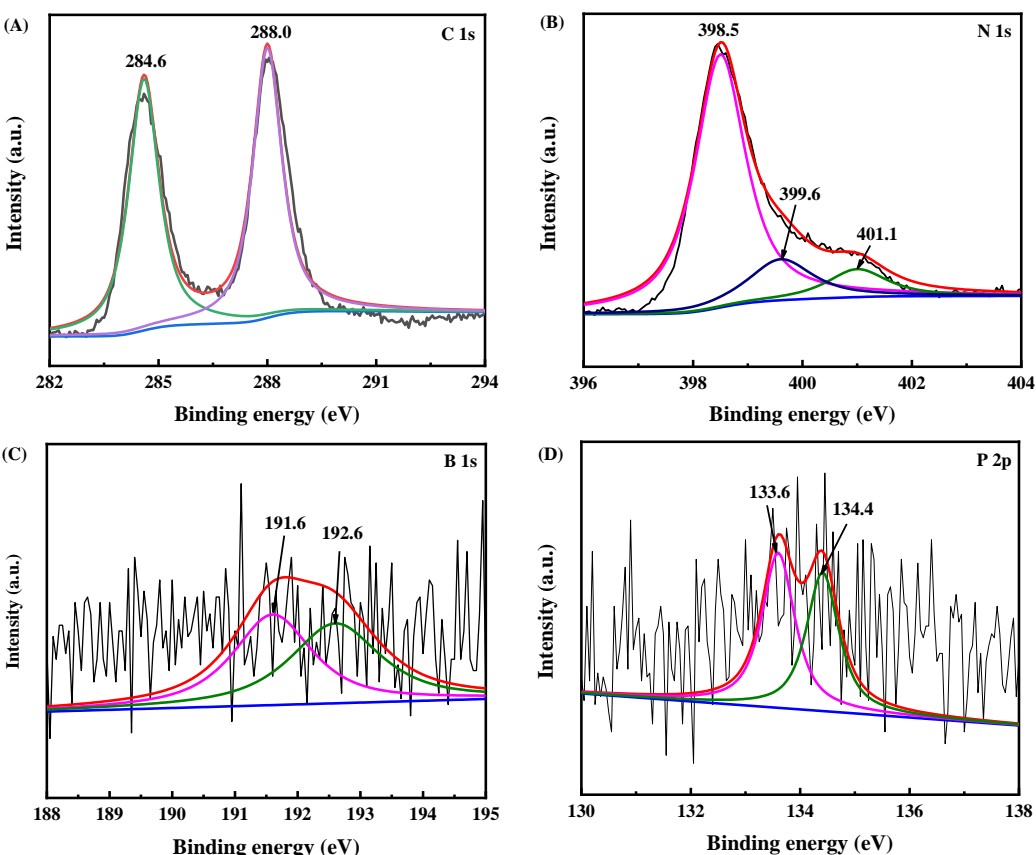

**Figure 3.** (**A**) C 1s; (**B**) N 1s; (**C**) B 1s; (**D**) P 2p XPS spectra of BP-CN-1.

$N_2$ adsorption–desorption was adopted to analyze the specific surface area ($S_{BET}$) of the synthesized catalysts and the $N_2$ adsorption–desorption curves and pore size distributions of catalysts were exhibited in Figure 4. The $S_{BET}$ value, BJH average pore diameter and ICP-AES analysis results of those catalysts are summarized in Table 1. The $S_{BET}$ of doped samples was larger than those of pure CN, indicating that the successful doping of B and P elements and larger $S_{BET}$ could provide more surface catalytic active sites [24]. Meanwhile, the $S_{BET}$ decrease observed in the doped samples might be caused by the increase in B and P dopings [34].

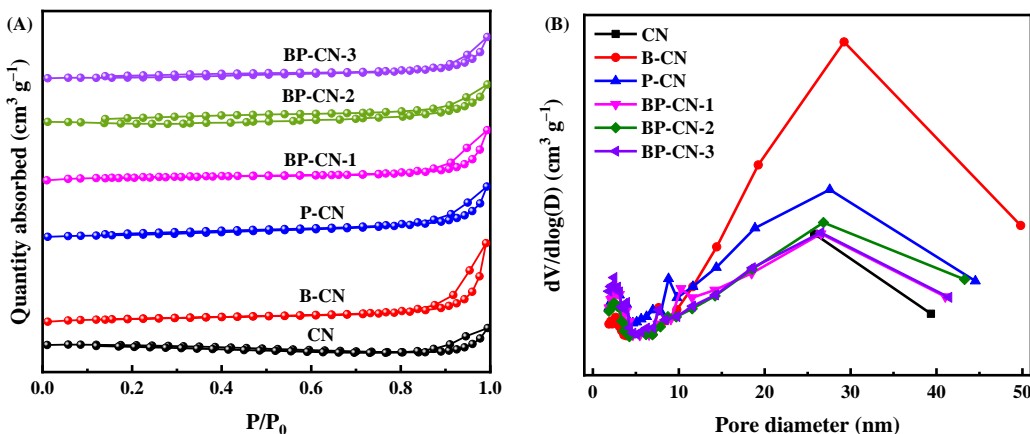

**Figure 4.** $N_2$ adsorption–desorption isotherms of the synthesized catalysts. (**A**) $N_2$ adsorption–desorption isotherms; (**B**) pore size distribution.

**Table 1.** $S_{BET}$, BJH average pore diameter and B, P mass fraction of catalysts.

| Samples | $S_{BET}$ (m$^2$ g$^{-1}$) | Average Pore Diameter (nm) | B (wt%) | P (wt%) |
|---------|------|------|------|------|
| CN | 2.2 | 0.01 | — | — |
| B-CN | 13.8 | 24.5 | 1.21 | — |
| P-CN | 12.5 | 14.1 | — | 1.27 |
| BP-CN-1 | 11.9 | 12.1 | 0.10 | 1.37 |
| BP-CN-2 | 9.3 | 13.7 | 0.48 | 3.56 |
| BP-CN-3 | 5.5 | 20.7 | 0.96 | 5.00 |

High-resolution transmission electron microscopy (HRTEM) and a scanning electron microscope (SEM) were adopted to reveal the morphology features of BP-CN-1 and the results are exhibited in Figure 5. TEM images (Figure 5A) clearly showed that BP-CN-1 was composed of large-size irregular sheets with some clear wrinkles, which was consistent with CN (Figure 5C). The SEM image (Figure 5B) revealed that BP-CN-1 possessed a rod structure; however, the SEM image of CN (Figure 5D) indicated irregular sheets in CN. This difference was caused by the addition of phosphorous acid [24].

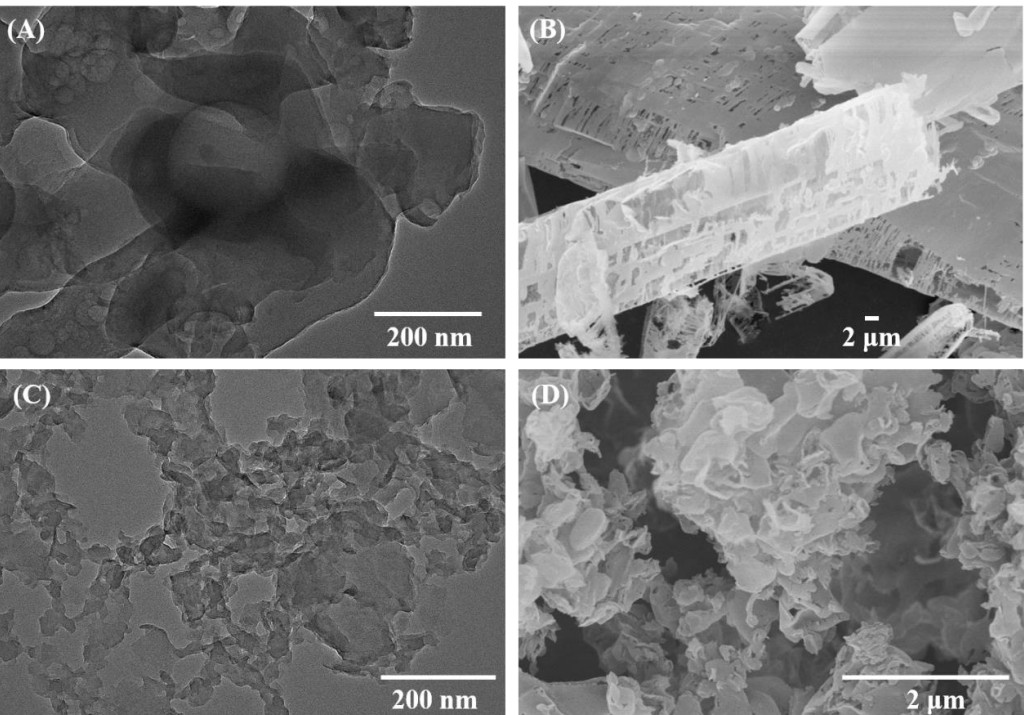

**Figure 5.** (**A**) HRTEM; (**B**) SEM images of BP-CN-1; (**C**) TEM; and (**D**) SEM images of CN.

### 3.2. Catalyst SCREENING

The catalytic activities of the synthesized catalysts were evaluated by the cycloaddition reaction of propylene oxide (PO) with $CO_2$ to form propylene carbonate (PC). All experimental results are listed in Table 2. No PC was obtained without a catalyst (Entry 1). Only a 2% PC yield was obtained with CN (Entry 2), and a 25% PC yield could be achieved with Bu$_4$NBr (Entry 3), which was generally applied as an effective co-catalyst in the literature [35,36]. When CN cooperated with Bu$_4$NBr, a 26% PC yield was obtained (Entry 4). The PC yield could be elevated when B- or P-doped CN acted as a catalyst with Bu$_4$NBr (Entries 5, 6). When B-CN/Bu$_4$NBr acted as a catalyst, a 50% PC yield was obtained (Entry 5), which might be caused by the enhancement of active sites of the CN surface after B dopings [21]. When P-CN/Bu$_4$NBr acted as a catalyst, a 44% PC yield was obtained (Entry 5), which might be ascribed to the electronic structure change in CN after P

dopings [23]. PC yield was obviously enhanced when BP-CN-n (n = 1, 2, 3) cooperated with $Bu_4NBr$ (Entries 7–9), especially BP-CN-1, where a 64% PC yield was obtained. However, the larger the doping content, the lower the catalytic activity of BP-CN-n. This phenomenon indicated that even B or P elements could provide a more catalytic active site but could not guarantee the reaction of the catalytic active site with PO. Herein, $S_{BET}$ played a key role in $CO_2$ cycloaddition reactions with the BP-CN-n/$Bu_4NBr$ catalyst [23]. Moreover, when the cocatalysts $Bu_4NBr$/$Bu_4NI$ were used in the cycloaddition reaction at 120 °C, 55% and 59% PC yield were obtained, respectively (Entries 10–11). Meanwhile, PC yield was 99% with BP-CN-1/$Bu_4NI$ at 120 °C, which was slightly higher than BP-CN-1/$Bu_4NBr$. However, $Bu_4NBr$ was more suitable than $Bu_4NI$ due to the economic point. Hence, BP-CN-1/$Bu_4NBr$ was employed as a catalyst to discuss the effects of reaction conditions.

**Table 2.** Catalytic performance of catalysts [a].

| Entry | Catalyst | CoCatalyst | Yield [b] (%) | Selectivity [b] (%) |
|---|---|---|---|---|
| 1 | — | — | 0 | — |
| 2 | CN | — | 2 | — |
| 3 | — | $Bu_4NBr$ | 25 | 98 |
| 4 | CN | $Bu_4NBr$ | 26 | 99 |
| 5 | B-CN | $Bu_4NBr$ | 50 | 99 |
| 6 | P-CN | $Bu_4NBr$ | 44 | 98 |
| 7 | BP-CN-1 | $Bu_4NBr$ | 64 | 98 |
| 8 | BP-CN-2 | $Bu_4NBr$ | 58 | 98 |
| 9 | BP-CN-3 | $Bu_4NBr$ | 53 | 98 |
| 10 [c] | — | $Bu_4NBr$ | 55 | 98 |
| 11 [c] | — | $Bu_4NI$ | 59 | 98 |
| 12 [c] | BP-CN-1 | $Bu_4NI$ | 99 | 98 |

[a] Reaction: PO 34.5 mmol, 100 °C, $CO_2$ 2 MPa, 6 h, 100 mg catalyst, 100 mg $Bu_4NBr$; [b] Yield and selectivity were determined by GC; [c] 120 °C.

### 3.3. Effects of Reaction Conditions

The effects of the reaction conditions were discussed and the results are exhibited in Figure 6. PC yield and reaction temperature showed a positive correlation (Figure 6A), and PC yield increased from 44% to 99% with the temperature increase from 90 to 130 °C due to the promotion of effective collisions among the catalyst and substrate [37]. When the reaction temperature was 120 °C, PC yield reached 95%. PC yield improved slightly when the temperature was further increased to 130 °C. Hence, 120 °C was regarded as a suitable reaction temperature. As depicted in Figure 6B, the PC yield increased when the $CO_2$ pressure increased from 0.5 MPa to 2.0 MPa; however, the PC yield showed a declining trend with higher $CO_2$ pressure. This might be ascribed to the fact that the high $CO_2$ pressure would dilute PO, resulting in a decrease in PC yield [38,39]. The effect of reaction time was also discussed (Figure 6C). PC yield could reach 95% within 6 h. Therefore, the reaction conditions were set as 120 °C, 2 MPa, and 6 h to investigate the recyclability and universality of BP-CN-1.

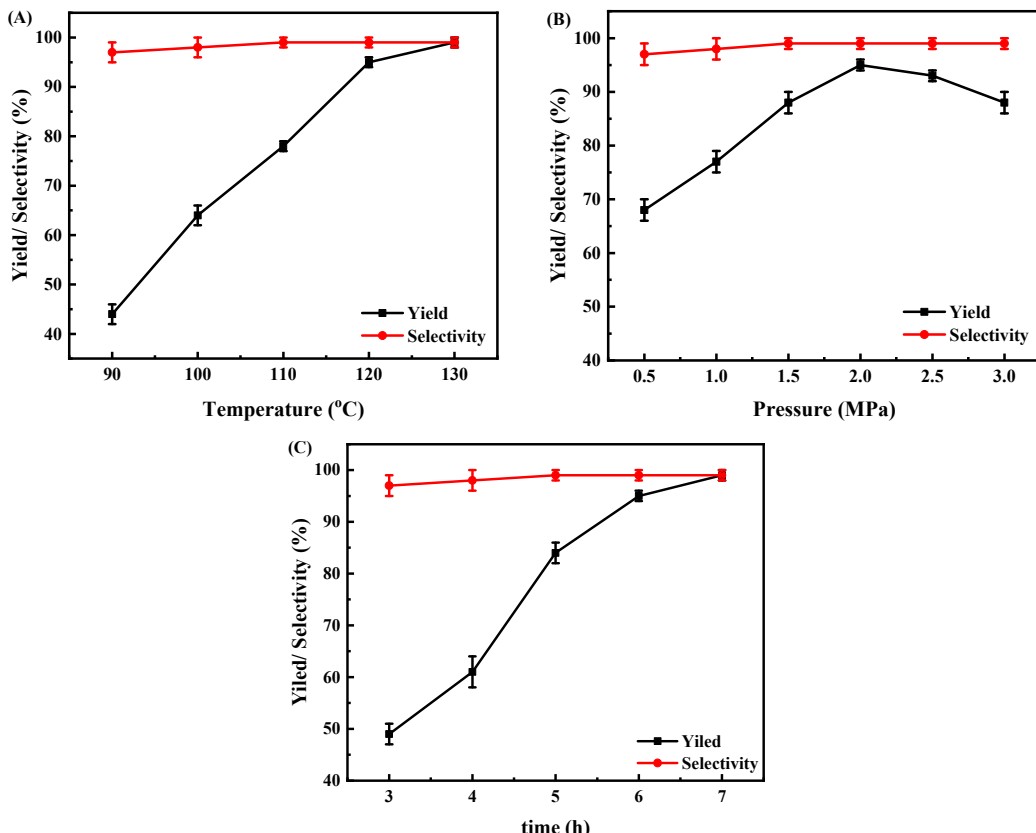

**Figure 6.** Effects of reaction parameters on $CO_2$ cycloaddition reactions over BP-CN-1. Reaction conditions: PO 34.5 mmol, 100 mg BP-CN-1, 100 mg Bu$_4$NBr, (**A**) effect of reaction temperature, 2.0 MPa, 6 h; (**B**) effect of $CO_2$ pressure, 120 °C, 6 h; (**C**) effect of reaction time, 120 °C, 2.0 MPa.

### 3.4. Catalyst Recyclability

Recyclability was a crucial evaluation standard. As exhibited in Figure 7A, BP-CN-1 possessed excellent recyclability and there was no PC yield loss during five recycling processes. In Figure 7B, there was no obvious difference between BP-CN-1 and reused BP-CN-1 in terms of FT-IR spectra. Additionally, the supernatant was separated and characterized by ICP-AES, and no B or P was detected. All this evidence shows the structural stability of BP-CN-1.

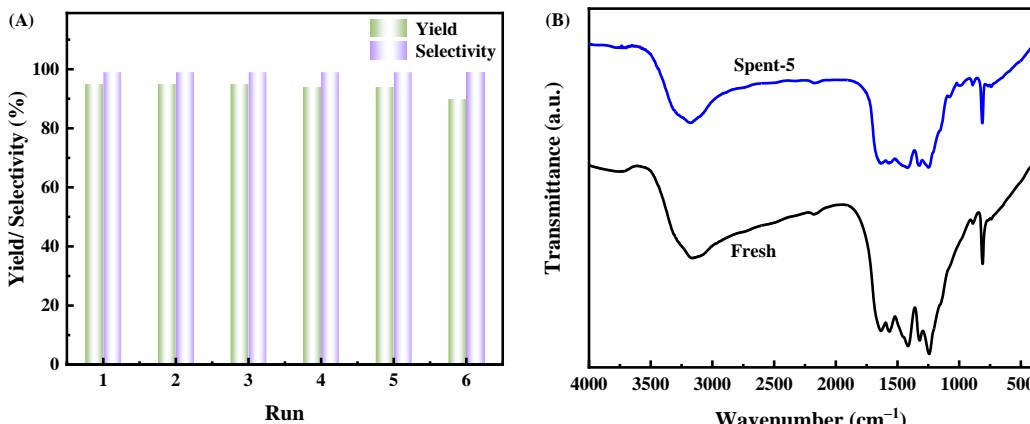

**Figure 7.** (**A**) Recyclability of BP-CN-1, conditions: PO 34.5 mmol, 100 mg BP-CN-1, 100 mg Bu$_4$NBr, 120 °C, $CO_2$ 2.0 MPa, 6 h; (**B**) FT-IR spectra of BP-CN-1 and reused BP-CN-1.

### 3.5. Catalyst Universality

The universality of BP-CN-1 was discussed and the results are shown in Table 3. The good yield and selectivity indicate that BP-CN-1 is adaptable to epoxides. 1,2-epoxybutane exhibited lower levels of conversion because of steric hindrance, preventing the nucleophilic attack of Br$^-$ on the C atoms of epoxides (Entries 1 and 2) [40,41]. A similar result was obtained for the butyl glycidyl ether (Entry 4). The conversion of epichlorohydrin was the highest (Entry 3), as this benefited from the electron-withdrawing effect of Cl$^-$. The electron-withdrawing effect promoted the cleavage of C−O [42]. The conversion levels of styrene oxide and cyclohexene were lower than PO, even with a longer reaction time (Entries 5 and 6). For styrene oxide, the benzene ring group prevented the bulky catalyst from approaching. Additionally, the effects of the electron donor from an aromatic group hindered the polarization of C−O in styrene oxide [43,44]. For cyclohexene, the low levels of conversion might be caused by the special two-ring structure, which led to the highest hindrance of the reactant [45,46].

**Table 3.** Universality of BP-CN-1 for catalytic conversion of various epoxides [a].

| Entry | Epoxides | Products | Results [a] | | |
|---|---|---|---|---|---|
| | | | Conversion [b] (%) | Yield [b] (%) | Selectivity [b] (%) |
| 1 | | | 96 | 95 | 99 |
| 2 | | | 87 | 86 | 99 |
| 3 | | | 99 | 99 | 99 |
| 4 | | | 74 | 73 | 98 |
| 5 [c] | | | 85 | 82 | 97 |
| 6 [c] | | | 66 | 63 | 96 |

[a] Reaction condition: PO 34.5 mmol, 120 °C, CO$_2$ 2 MPa, 6 h, 100 mg BP-CN-1, 100 mg Bu$_4$NBr; [b] Conversion and selectivity were determined by GC; [c] 12 h.

### 3.6. Comparisons of CN-Based Catalyst

The comparisons between the catalytic performance of B, P-CN-1 with various CN-based materials for the cycloaddition of $CO_2$ to epoxides are listed in Table 4. Overall, a high temperature (>100 °C) was an essential condition to obtain an excellent yield. N, N-dimethylformamide (DMF) was added to the catalytic systems to promote the cyclic carbonate yield (Entries 1–3). Concerning single B-doped CN (Entries 4–5), not only did the temperature need to reach 130 °C, but it also took a long time to reach an outstanding yield. For the dual-doped K, B-CN-4 catalyst, a lower temperature (110 °C) was needed due to the metal K, which accelerated the epoxides' ring-opening (Entry 6). The HCN-CN catalyst obtained more –COOH and –OH groups, leading to a lower temperature (Entry 7). The P-CN-2 catalyst sacrificed a lower temperature and needed less time to achieve an almost 100% conversion due to the change in its electronic structure (Entry 8). As a result, the B, P-CN-1 catalyst showed competitive activity in the $CO_2$ conversion field with epoxides; hence, the B, P-CN-1 catalyst could potentially be utilized in $CO_2$ cycloaddition reactions. To compare the activity of different modified CN catalysts under various conditions, the TOF and TON values were supplemented, which were calculated on the mass of produced cyclic carbonate per gram of catalyst per hour according to the literature [20] because the other modified CN catalysts could not provide the amounts of active sites. As a result, BP-CN-1 showed competitive activity in the $CO_2$ conversion field with epoxides.

**Table 4.** Comparisons of B, P-CN-1 catalyst with reported CN catalysts in the $CO_2$ cycloaddition with epoxides [a].

| Entry | Catalysts | Solvent | Reaction Condition | | | Yield (%) | TOF | TON | Reference |
| --- | --- | --- | --- | --- | --- | --- | --- | --- | --- |
| | | | Tem. (°C) | Press (MPa) | T(h) | | | | |
| 1 | eg-CN-OH | DMF | 140 | 2.0 | 6.0 | 60 | 5.8 | 35.0 | [47] |
| 2 | mpg-CN/ZnCl$_2$(10) | DMF | 140 | 2.5 | 6.0 | 72 | 7.0 | 42.0 | [48] |
| 3 | Zn-CN/C | DMF | 140 | 2.0 | 8.0 | 74 | 5.4 | 43.3 | [49] |
| 4 [b] | B$_{0.1}$CN/SBA-15 | – | 130 | 3.0 | 24 | 95 | 1.2 | 28.5 | [22] |
| 5 | MCNB(0.01) | – | 130 | 2.8 | 6.0 | 31 | 1.5 | 9.1 | [21] |
| 6 | K,B-CN-4(Bu$_4$NBr) | – | 110 | 2.0 | 6.0 | 87 | 5.1 | 30.7 | [31] |
| 7 | HCN-FD(Bu$_4$NI) | – | 110 | 2.0 | 6.0 | 92 | 5.4 | 32.4 | [50] |
| 8 | P-CN-2 (Bu$_4$NBr) | – | 100 | 2.0 | 4.0 | 99 | 4.8 | 19.3 | [23] |
| 9 | B,P-CN-1(Bu$_4$NBr) | – | 120 | 2.0 | 6.0 | 95 | 5.5 | 33.1 | This work |

[a] PO; [b] SO; TOF: the mass of produced PC per gram catalyst per hour; TON: the mass of produced PC per gram catalyst.

### 4. Conclusions

In conclusion, three kinds of B and P co-doped carbon nitrides (BP-CN-n, n = 1, 2, 3) were synthesized, characterized and applied in $CO_2$ cycloaddition reactions with the co-catalyst Bu$_4$NBr. BP-CN-1/Bu$_4$NBr exhibited the best catalytic activity because BP-CN-1 possessed the largest specific surface area and reasonable B and P doping contents. BP-CN-1/Bu$_4$NBr could promote the cycloaddition reaction of $CO_2$ with propylene oxide under 120 °C, 2 MPa, 6 h, without the employment of metal or solvents. BP-CN-1/Bu$_4$NBr also possessed remarkable recyclability and universality. Hence, BP-CN-1/Bu$_4$NBr could be an environmentally friendly catalyst for $CO_2$ cycloaddition reactions in the future. Moreover, in subsequent work, the nucleophilic group could be introduced into the catalyst to avoid the application of a cocatalyst and the $CO_2$ cycloaddition reaction could be finished under mild conditions.

**Supplementary Materials:** The following supporting information can be downloaded at: https://www.mdpi.com/article/10.3390/catal12101196/s1, Figure S1: The GC-MS of (A) 1,2-epoxybutane (BO), (B) epichlorohydrin (ECH), (C) butyl glycidyl ether (BGE), (D) styrene oxide (SO) and (E) cyclohexene (CHO) before and after reaction.

**Author Contributions:** Data curation, C.Y.; formal analysis, T.Y.; investigation, C.Y.; resources, X.Z.; writing—review and editing, C.Y. All authors have read and agreed to the published version of the manuscript.

**Funding:** This work was supported by the Civil Aviation Development Fund Education Talents Project (0252101), the Key Project of Civil Aviation Flight University of China (ZJ2020-06) and the general fund of Civil Aviation Flight University of China (BSJ2021-2, BSJ2021-3, BSJ2022-6).

**Conflicts of Interest:** The authors declare no conflict of interest.

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
