# Peer review of "Boron and Phosphorus Co-Doped Graphitic Carbon Nitride Cooperate with Bu4NBr as Binary Heterogeneous Catalysts for the Cycloaddition of CO2 to Epoxides"

_catalysts, doi:10.3390/catal12101196_

Round 1

Reviewer 1 Report

In this manuscript, the authors report about dual boron and phosphorus co-doping graphitic carbon nitride (BP-CN)/TBAB catalytic system for the CO2 cycloaddition with epoxides. Although the manuscript is well-written and sounds interesting, there is some questions which have to be addressed before publication.

1) I recommend to provide the information about BP-CN-1, BP-CN-2, BP-CN-3 (the synthesis and how their obtained and the difference between them) in the main text. Otherwise, it is difficult to understand for the readers.

2) There is no big difference between the activity of TBAB and the mixed system BP-CN-1/TBAB (Table 2). And looking on Figure 5, 120 oC is the optimal temperature. So, how about to perform the reaction with TBAB at 120 oC? Moreover, it would be interesting to see the result at the optimal conditions with TBAI (tetrabutyl ammonium iodide) to see the benefit of the here applied catalytic system.

3) The authors have to provide the yields of the products instead of the conversion in Table 3. Moreover, it would be highly desirable to provide the 1H NMR spectra to check the purity of the products.

4) It is not clear, does a new batch of TBAB was added or no after recycling?5) Table 4. I recommend to provide TON and TOF values to compare the results with literature data?

6) Page 3. Exp. Section: catalytic activity test-first sentence - “stainless steel autoclave stainless-steel autoclave“ is double times.

7) The recent papers on CO2 fixation into cyclic carbonates can be cited as well:

Chem. Eur. J. 2022, 28, e202200622

Inorg. Chem. Front. 2022, 9, 2969-2979

Asian J. Org. Chem. 2022, 11, e202100811

J. CO2 Utilization 2022, 57, 101884

Russ. Chem. Bull. 2021, 70, 1324-1327

Inorg. Chem. Front. 2021, 8, 3871–3884

Author Response

We would like to thank the editor and the reviewers for the comments on our manuscript entitled “Boron and phosphorus co-doped graphitic carbon nitride cooperate with Bu4NBr as binary heterogeneous catalysts for the cycloaddition of CO2 to epoxides” (catalysts-1942435), these comments are valuable and helpful for improving our paper. We have studied the comments carefully to improve the manuscript, and we hope it will meet with their approval. Meanwhile, the English have been improved by MDPI English Editing. All revisions to the manuscript are marked up using the “Track Changes” function.

Reviewer 2 Report

In this manuscript “Boron and phosphorus co-doped graphitic carbon nitride cooperate with Bu4NBr as binary heterogeneous catalysts for the cycloaddition of CO2 to epoxides”, three boron, phosphorous co-doping graphitic carbon nitride were prepared, and characterized. The catalytic activity in CO2 and epoxide cycloaddition was tested with the presence of Bu4NBr. And the 

catalyst system was adapted to up to 6 epoxides. There is a good amount of work. 

However, further manuscript polishing is needed. For example, on page 1, the abstract: “In this manuscript, a series of boron and phosphorus…” that’s not precise. Please delete “a series of” or replace it with “three”. “Moreover, BP-CN-1/Bu4NBr catalysts exhibited excellent substrate versatility to various epoxide…” with “Moreover, BP-CN-1/Bu4NBr catalysts system is compatible with various epoxide…” would be better.

Regarding the experiment procedure (page 2), the molar amount of each chemical should be given. The typical format should be “MA (5.0 g, 39.7 mmol)”. “…with distilled water to remove the remaining H+…”, it is arguable. Not only H+ could be washed away by water, but also hydrogen phosphite, MA salt, and even boric acid should be soluble in water. “Catalytic activity test” (page 3), and “stainless-steel autoclave” should be deleted. 

As to catalyst characterization (page 3), FT-IR, XRD, XPS, and N2 adsorption-desorption were used to characterize the catalyst synthesized. Fine work has been done. However, more effort should be done to try to find the clue of a new bond formed (e.g., N-B, N-P) in the interpretation of the spectra. Page 4, “standard carbon” is not a scientific term. “…indicating that P replaces C in triazine rings…” that doesn’t make sense. Please be more specific. Page 7, “Figure S1A”, and “Figure S1B” cannot be found.

For the application of the catalyst system, selectivity is a significant parameter. The data was depicted in almost every table and figure. However, there is no explanation of what the term stands for. Table 2, “Yield” was used. Table 3, “Conversion” was used. The author claimed they were all determined by GC. Please explain how. How about the stability of the epoxide in GC? Regarding the catalyst recyclability, the way the author did the catalyst recycling should be mentioned in the context. If the catalyst keeps at high activity after 5 recycles, more experiments could be done.

Author Response

(The authors gave the same response as above.)

Reviewer 3 Report

Yang et al present a good paper towards cloaddition entitled as Boron and phosphorus co-doped graphitic carbon nitride cooperate with Bu4NBr as binary heterogeneous catalysts for the cycloaddition of CO2 to epoxides

This paper will be a usueful entry towards readers and forthcoming sxientist to work on this and i recommend this paper suitable for publication after some minor concerns 

1. Abstract need to be modified 

2. Author have to remove all typo errors in manuscript

3. In introduction please cite your study with latest reference such as https://doi.org/10.1016/j.mattod.2022.04.002

4. Explain future prospective of the study 

5. Why author use this material. Scope and novelity should be explained 

Author Response

(The authors gave the same response as above.)

Round 2

Reviewer 1 Report

The authors have carefully revised the manuscript taking into consideration the comments of the reviewers. I’m bit disappointed by this answer “Exactly, 1H NMR spectra is a more accurate method than GC, but it can’t be achieved because of the influence of COVID-19. And the test details of GC were also given in the Catalytic activity test of the revised manuscript to prove the results were reliable.” So, my question is how the authors could prove the formation of cyclic carbonate products without NMR? Furthermore, I also recommend to add the next information “Additionally, the PC yield was 55% for TBAB at 120 oC, this data was obtained in our experimental process before. When the cocatalyst was replaced by TBAI under the optimal conditions, 99% PC yield was received, which was slightly higher than TBAB” into main text.

Author Response

We would like to thank the reviewers for the comments on our manuscript entitled “Boron and phosphorus co-doped graphitic carbon nitride cooperate with Bu4NBr as binary heterogeneous catalysts for the cycloaddition of CO2 to epoxides” (catalysts-1942435), these comments are valuable and helpful for improving our paper. We have studied the comments carefully to improve the manuscript, and we hope it will meet with their approval. All revisions to the manuscript are marked up using the “Track Changes” function.

Reviewer 2 Report

Thanks for the author’s cover letter and the carefully addressed the raised concerns. Most of them have been well handled. However, there are a few that still need to be further taken care of.

1)    Keywords: please keep the first letter be upper case.

2)    Legend of tables: the conditions should be CO2 (2 MPa).

3)    Please redraw the chemical structures in “Table 3”, so that it looks not so casual.

4)    Recyclability: If possible, please include the 6th recycled data in “Figure 7”.

Author Response

(The authors gave the same response as above.)

Round 3

Reviewer 1 Report

The authors have addressed all comments; so, the paper is publishable now.